

# A class of probability distributions for application to non-negative annual maxima

Earl Bardsley[1]

[1] School of Science, University of Waikato, Hamilton 3240, New Zealand

*Correspondence to*: Earl Bardsley (earl.bardsley@waikato.ac.nz)

**Abstract.** Many environmental variables of interest as potential hazards take on only positive values, such a wind speed or river discharge. While recognising that primary interest is for largest extremes, it is desirable that distributions of maxima for design purposes should be subject to similar bounds as the physical variable concerned. A modified univariate extreme value argument defines a set of distributions, all bounded below at zero, with potential for application to annual maxima. Let $f(x)$
be a probability distribution over the range, $0 \leq x \leq \omega$, where $0 < \omega \leq \infty$. Define $X_* = \max(X_1, X_2, .. X_N)$ to be the maximum value of a random sample of size $N$ drawn from $f(x)$. Also, define the transformation $Y_i = g(X_i)$ where $g(X)$ is any positive monotonically decreasing function of $X$. This would include, for example, $Y = X^{-1}$ but not $Y = -X$. Because the $Y_i$ are independent random variables bounded from below at some non-negative value $\varepsilon$, it follows from extreme value theory for minima that for sufficiently large $N$ the random variable $Y_* = \min(Y_1, Y_2, .. Y_N)$ will follow a Weibull distribution with
cumulative distribution function:

$$F(y) = pr(Y_* \leq y) = 1 - \exp\{-[(y - \varepsilon)/\alpha]^c\} \qquad \varepsilon \geq 0, \alpha > 0, c > 0$$

where $\varepsilon = g(\omega)$ and $\varepsilon$, $\alpha$, and $c$ are respectively location, scale, and shape parameters. The distribution $F(y)$ holds generally as an extreme value expression for sufficiently large $N$, irrespective of which of the three possible asymptotic extreme value distributions of sample maxima holds for $X_*$. Therefore, the limit Weibull distribution for, say, $Y_* = X_*^{-1}$ has no less validity
as a single expression for obtaining exceedance probabilities than the generalized extreme value distribution applied directly to $X_*$. If follows that a class of probability distributions for possible use with positive-valued annual maxima can be defined from the application of the inverse function $g^{-1}$ to Weibull random variables for $\varepsilon \geq 0$. All distributions so obtained are defined over the range $0 \leq x \leq \omega$, which actually excludes all of the asymptotic extreme value distributions of maxima except for the special case of the Type 2 extreme value distribution with location parameter at zero. It is to be expected, however,
that the asymptotic distributions will sometimes hold to a high level of approximation within the 0, $\omega$ interval. No specific distribution is advocated for annual maxima application because concern here is only with drawing attention to the existence of the distribution class. The transformation approach is illustrated with respect the distribution of reciprocals of random variables generated from a 3-parameter Weibull distribution with $\varepsilon \geq 0$.



## 1 Introduction

Jenkinson (1955) introduced the univariate generalized extreme value distribution (GEV) to analysis of environmental maxima or minima, subsequent to its original mathematical formulation by von Mises (1936). The GEV distribution for largest extremes is a single expression which incorporates the three asymptotic extreme value distributions of sample
maxima derived by early workers in the field (Fisher and Tippett, 1928; Gnedenko, 1943). An historical overview is given by Kotz and Nadarajah (2000). Because of its natural linkage to maxima, the GEV distribution has been applied for design purposes in numerous instances with respect to annual maxima such as flood discharges, wave heights, and wind speeds (Coles, 2001). Focus for maxima is of course on distribution upper tails but it is more reflective of reality if probability distributions for design purposes are defined within the same bounds as the physical variable concerned, noting that many
environmental variables with hazard potential such as wind speed and river discharge are bounded below at zero. In this regard the Type 3 and Gumbel asymptotic distributions of maxima both extend into the negative domain, albeit with very small probability in practical applications.

This brief communication makes an alternative extreme value argument leading to a class of distributions, all bounded below
at zero, which could have potential application to annual maxima and with no less theoretical justification than the GEV. There is no data-based argument made for any one distribution but the approach is illustrated for the particular case of reciprocals of 3-parameter Weibull random variables.

## 2. Alternative distribution class

Let $f(x)$ be an unknown probability distribution defined over the range, $0 \leq x \leq \omega$, where $0 < \omega \leq \infty$. Define $X_* = \max (X_1,$
$X_2, .. X_N)$ to be the maximum value of a random sample of size $N$ drawn from $f(x)$. Also, define the transformation $Y_i = g(X_i)$ where $g(X)$ is any positive monotonically decreasing function of $X$. This would include, for example, $Y = X^{-1}$ but not $Y = -X$. Because the $Y_i$ are independent random variables bounded from below at some non-negative value $\varepsilon$, it follows from extreme value theory for minima that for sufficiently large $N$ the random variable $Y_* = \min (Y_1, Y_2, .. Y_N)$ will follow a Weibull distribution with cumulative distribution function:


$$F(y) = pr(Y_* \leq y) = 1 - \exp\{-[(y - \varepsilon) / \alpha]^c\} \qquad \varepsilon \geq 0, \alpha > 0, c > 0 \qquad (1)$$

where $\varepsilon = g(\omega)$ and $\varepsilon$, $\alpha$, and $c$ are respectively location, scale, and shape parameters.

The distribution $F(y)$ holds generally as a single extreme value expression for sufficiently large $N$, irrespective of which of
the three possible asymptotic extreme value distributions of largest extremes holds for $X_*$. Therefore, the limit Weibull distribution for, say, $Y_* = X_*^{-1}$ has no less validity as a general expression for obtaining exceedance probabilities than the generalized extreme value distribution applied to $X_*$.





If follows that a class of probability distributions for possible use with positive-valued annual maxima (taking a "year" as a random sample) can be defined from the application of the inverse function $g^{-1}$ to random variables generated from a three-parameter Weibull distribution with $\varepsilon \geq 0$. All the distributions so obtained are defined over the range $0 \leq x \leq \omega$, which

excludes all of the asymptotic extreme value distributions of maxima except for the special case of the Type 2 extreme value distribution with location parameter at zero. It is to be expected, however, that the asymptotic distributions will sometimes hold to a high level of approximation within the 0, ω interval.

Each permissible inverse function transformation of Weibull random variables will define a different distribution in the class. The transformation approach is illustrated in the next section, with respect to the particular case of the distribution of

reciprocals of random variables generated from a 3-parameter Weibull distribution with $\varepsilon \geq 0$.

## 3. Illustration (Weibull reciprocal transformation)

For $N$ sufficiently large $Y_* = X_*^{-1}$ will follow a Weibull distribution as defined by Eq. (1) and the inverse transformation of the Weibull random variable $W$ is given by $Z = W^{-1}$. The resulting distribution of $Z$ is referenced here for convenience as the H distribution, with cumulative distribution function and probability density function respectively given by:


$$H(x) = pr(Z \leq x) = \exp[-\beta^c (x^{-1} - \omega^{-1})^c] \qquad (2)$$

$$h(x) = \exp[-\beta^c (x^{-1} - \omega^{-1})^c] \beta^{c-2} (x^{-1} - \omega^{-1})^{c-1} (\beta x^{-1})^2 c \qquad (3)$$

$\omega > 0, \quad \beta > 0, \quad c > 0, \quad 0 < x < \omega$

where ω is the upper bound to the distribution range and β and $c$ are scale and shape parameters, respectively.

For $\omega \to \infty$ the H distribution tends toward a Type 2 distribution of largest extremes with location parameter at zero. Otherwise the distribution is bounded above at some finite value ω, which corresponds to $\varepsilon^{-1}$ in Eq. (1).


Bounded H distributions can be plotted in standard form by setting ω = 1 (Fig. 1). As expected, there can be similarity between bounded H distributions and extreme value distributions. In particular, plot $b$ in Fig. 1 is almost identical to a Gumbel probability density function with location parameter 0.5 and scale parameter 0.05. However, there is a difference in that the H distribution here is bounded both above and below while Gumbel distributions are always unbounded in both

directions.





The H distribution can also have some forms different to any of the three asymptotic extreme value distributions. A convenient illustration is to plot the H distribution on a Gumbel plot ($x$ on the vertical axis with Gumbel $y$ on the horizontal axis, where $y = -\ln\{-\ln[H(x)]\}$ ). In terms of the symbolism for the H distribution used here, this relation of $x$ as a function of $y$ is given by:

$$x = [\omega^{-1} + \beta^{-1} \exp(-y/c)]^{-1} \qquad (4)$$

Unlike the corresponding extreme value expressions on Gumbel plots, Eq. (4) can include an inflection point (Fig. 2). From the purely pragmatic viewpoint it is therefore possible that the H distribution will sometimes give a better match to annual maxima than the generalized extreme value distribution.

## 3. Discussion

The H distribution may or may not find use in the analysis of annual maxima for design purposes. The distribution does have the attraction of permitting both upper and lower bounds, consistent with presence the physical upper bounds which must apply to all environmental variables. However, as noted earlier, there are any number of $g^{-1}$ Weibull transformations which might be envisaged and no particular member of the resulting distribution class is advocated here.

With respect to extreme value theory, no argument that can be made for some g(.) transformation being superior in the sense that increasing $N$ will in general result in the distribution of $Y_*$ converging more quickly to the Weibull limit distribution of smallest extremes, as compared to the rate at which the distribution of $X_*$ converges to the GEV distribution of maxima. It will always be possible to find situations of faster and slower convergence rates. For example, the distribution of sample maxima from an exponential distribution converges quickly to the limit Gumbel distribution (Bury, 1975). This can be compared with the distribution of reciprocals of the same sample maxima converging more slowly to the limit Weibull distribution of minima. On the other hand, if the $X_i$ are absolute values of standard normal random variables then the distribution of $Y_* = \exp(-X_*^{3/2})$ converges to the Weibull limit more quickly than the distribution of $X_*$ converges to its Gumbel limit as $N$ increases. In reality the distribution of the $X_i$ values is always unknown so comparisons of convergence rates for specified distributions for $X_i$ are not very helpful.

One obvious restriction is that the choice of transformation must be independent of any given $X_*$ data set, because optimising to achieve the best Weibull data fit to $Y_*$ becomes simply an exercise in fitting a flexible distribution.





Application of any of the distributions derived from Weibull transformations will need to consider the usual issues of estimation error. In particular, any upper bound obtained from data fitting is best regarded more as an artefact of the distribution concerned than an estimate of a physical limit. Examples of fitting to annual maxima are deliberately avoided here because fitting to a few data sets will not establish the general utility of any distribution. Superiority of any one annual

maxima distribution can never be established by comparison with data histograms, but it could be useful nonetheless to demonstrate via extensive data application that one or more members of the new distribution class are at least no less applicable for data description than existing distributions for annual maxima like the GEV. It is hoped that this brief communication may encourage some investigations of this type.

## 4 Conclusion

An argument has been made for the existence of a set of distributions applicable to annual maxima defined over the same physical range as the environmental variable under consideration. While these distributions are not extreme value distributions in themselves they have support from extreme value theory through their linkage to the Weibull distribution, which is the asymptotic distribution of minima of transformed values of the variable concerned.

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





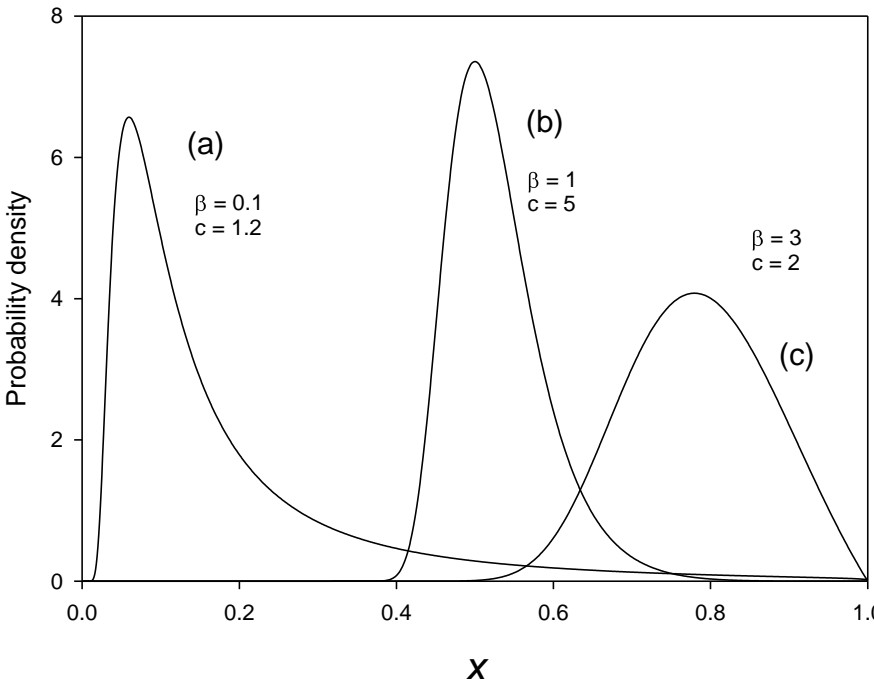

**Figure 1: Selected bounded H distributions in standard form for various parameter combinations. Plots (a), (b), and (c) respectively have similarities with Type 2, Gumbel, and Type 3 extreme value distributions.**

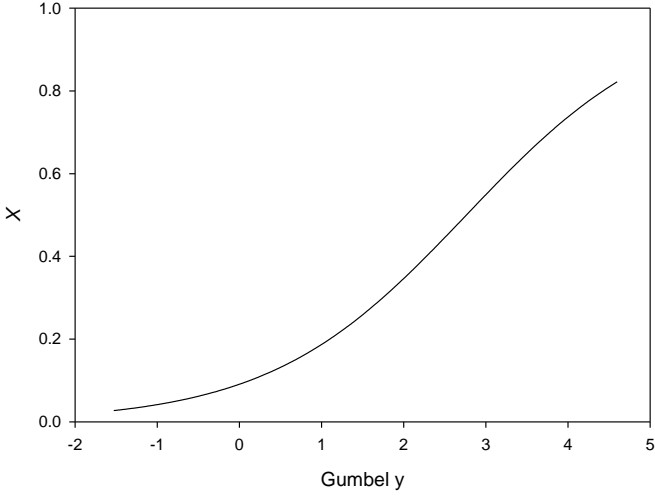

**Figure 2: Gumbel plot corresponding to the H distribution of plot (a) in Fig. 1, extending over the range H(0.01) to H(0.99), corresponding to Gumbel $y$ = - 1.53 to 4.60, respectively.**