# Peer review of "A class of probability distributions for application to non-negative annual maxima"

_Hydrology and Earth System Sciences, 2017_

## Referee Comment (RC1) · Anonymous Referee #1 · 8 Jun 2017

[british]article [T1]fontenc [utf8]inputenc amsmath amsthm

babel

**General comments**

The approximation of the distribution of extremes is of great interest for both hydrology and water management. Given the gap between theory and practice it is always a plea-sure to see a somewhat more theoretical approach. The examination of alternatives to the standard GEV is welcome as well, it is known that GEV does not cover all possible ways to examine distributions of extremes, but theoretical alternatives do not seem to

get much attention in hydrology.

The paper under review tries to provide one such alternative. It takes a random sample of $n$ elements from a random variable. The random variable has range $[0, \omega]$ with $0 < \omega \le \infty$. It defines the new random variable

$$X^* = \max(X_1, X_2, \ldots, X_N)$$

Moreover it selects a positive monotonically decreasing function $g$ and defines new random variables $Y_i = g(X_i)$ and defines

$$Y^* = \min(Y_1, Y_2, \ldots, Y_n)$$

It then assumes that for $Y^*$ a limiting distribution exists. It takes a variable $W$ distributed according to that limiting distribution and defines a random variable $Z = g^{-1}(W)$. It then claims that the distribution of $Z$ can be used to approximate that of $X^*$ and that this has no less validity than the use of an extreme value distribution. Along the way several assumptions are made, mostly implicitly. As far as I can see, some of these do not hold in the full generality needed in the paper in its current form and therefore it does not yet show that the method "has no less validity than the use of an extreme value distribution".

**Specific comments**

I have three main comments.

- The text assumes that a limiting distribution exists for $Y^*$. However, not all distributions are in a domain of attraction of an extreme value distribution, there are examples of distributions that are not in a domain of attraction of an extreme value distribution.

[Figure]

- Even if $X^*$ does have a limiting distribution, becaue of the potential non-linearity of $g$, this not necessarily imply that $Y^*$ has a limiting distribution.

- The text assumes that $Y^*$ has the Weibull distribution because the underlying random variable is bounded from below. This is not necesarily true, even for distributions that are in the domain of attraction of one of the extreme value dsitrbutions for minima. There are variables that are bounded from below where the limit is in the domain of attraction of the variation of Gumbel that applies to minima.

**Technical comments**

The comments refer to the pages (P) and lines (L) in the pdf as downloaded from the website.

**P2 L8-12.** The claim is made that "... it is more reflective of reality if probability distributions for design purposes are defined within the same bounds as the physical variable concerned ...". This is similar to an argument sometimes brought against the normal distribution for means of sums of random variables. In both cases one might use the counter argument that these are limit distributions as the number of random variables goes to infinity. The mathematical theory behind the limit process makes clear that, given the choice of limit process and the assumptions made, these limit distributions are the only ones that are "reflective of reality".

**P2 L19.** Lower case $f$ is usually used to refer to a probability density function (pdf), not a (cumulative) distribution function (cdf). If there is a pdf then it implies that the cumulative distribution function is absolutely continuous and life becomes much easier. Amongst other things this rules out some pathological behaviour for the limit process that leads to the extreme value distributions.

**P2 L19-20.** Not all definitions of a random sample include independence. Do you assume that that the $X_i$ are independent?

**P2 L21.** Two comments.

1. Not every monotonic function is everywhere continuous. To avoid unnecessary complications one might add the condition that $g$ is continuous.

2. A note stating that monotonicity of $g$ implies that if $X$ is a random variable then $g(X)$ is a random variable might help readers.

**P3 L2** The inverse function $g^{-1}$ exists only if $g$ is strictly monotonic on $(0, \omega)$. Usually the following definitions are used: a function $f$ is monotonic decreasing when $x < y \Rightarrow g(x) \geq g(y)$ and strictly monotonic decreasing when $x < y \Rightarrow g(x) > g(y)$.

**P3 L3** If $Y^*$ has a Weibull distribution then it is not guaranteed that $g^{-1}$ is defined on the whole of the range of $Y^*$ which always runs up to $+\infty$. Take for instance

$$g(x) = \{\, 3 - \frac{1}{1+|x|} x < 0 2 x = 0 1 + \frac{1}{1+x} x > 0$$

which has a range that lies in $[1, 3]$.

---

## Short Comment (SC1) · 12 Jul 2017

As this is the third time I see this paper under review (after Advances in Water Resources and Envoronmental Modelling Software), I can conclude that the Author strongly believes in his message. I have only some very general comments that can help to put the material in the context, at least. Leaving aside the specific (monotonic decreasing) transformation, the duality between maxima and minima and their distribution is not new at all.

For example, it is well known that the Weibull distribution is a reverse (reflected to the origin) GEV and their parameters are linked by analytical relationships (see Hosking and Wallis 1997, p. 195, as well as older papers and books going back to the

1970's). This property (which actually corresponds to switching maxima and minima) is exploited for instance in the fitting procedures implemented in 'lmom' and 'lmomco' R packages.

Concerning the reciprocal transformation, the Frechet distribution (for the largest order statistics) is also known as inverse Weibull. Actually, the 'new' distribution H in Eq. 2 is just a specific parameterization of one of the uncountable generalizations of the inverse Weibull (aka Frechet, aka Log-Gompertz, etc.) distribution.

Therefore, as specified in my previous reports, the note under review does not seem to provide any new insight. On the other hand, I think that proposing 'new' statistical results in hydrologic journals is not a good strategy to guarantee the quality of the scientific production, as there is a very good chance that professional statisticians with expertise in EVT do not comment on the paper. Of course, there are examples of distributions (or fundamental stochastic results) introduced in in hydrological journals, such as the Kumaraswami distributions or some fundamental results from Benoit Mandelbrot on scaling. However, in those cases, the Authors used a completely different level of detail and accuracy compared with the present note. As mentioned by the Referee 1, the Author's proposal implies a number of assumptions that are not discussed at all, overlooking lots of theory on EV and distributions.

Finally, even though there are many good books on EVT, I strongly recommend to everyone approaching EVT a careful reading of Emil Julius Gumbel's benchmark 'Statistics of Extremes', which is dated but better than many others and can help avoiding many (many) problems in this field.

Sincerely

Francesco Serinaldi

References

Gumbel E.J. (1958) Statistics of extremes, Columbia University Press, 1958 (reprinted

by Courier Corporation, 2004; Dover books on mathematics)

Hosking J.R.M., Wallis J.R. (1997) Regional Frequency Analysis: An Approach Based on L-Moments, Cambridge University Press
* * *

---

## Short Comment (SC2) · 13 Jul 2017

Dear Earl,

I'm really sorry that you considered my comments as 'straw man tactics', since there is no tactics in my remarks. What advantages would I get from this public discussion? Actually, I would be quite foolish if I used 'straw man tactics' in an eponymous report. Fortunately, we are old enough to know that 'straw man tactics' along with 'conflict of interest' and other similar arguments are only stratagems used in our business.

By the way, in this case, I'm not a reviewer but just a commenter, so you can be sure that the review process will proceed in a transparent way. Moreover, the handling Editor has enough experience and expertise in statistics and hydrology to make a fair

and correct decision, filtering out my 'straw man' remarks.

Coming to technical stuff, I'm happy that you agree with me about the transformation $Y = 1/X$. In fact, in my comment, I only stressed that both $Y = 1/X$ and $Y=-X$ provide a switch between maxima and minima, and both are well known and already exploited, resulting in distributions already discussed in the literature and widely used. I mentioned $Y=-X$ to highlight that also this transformation yields known results (even if I recognized that you did not consider it) because both $Y = 1/X$ and $Y=-X$ follow the same rationale with the same purpose. So, my question is very simple: do we really need a paper/technical note saying that 'for a Weibull random variable, $Y = 1/X$ yields a Frechet/Inverse Weibull (...or any other of the names used along the years for this family) and $Y=-X$ (if you want to add this case) yields a GEV'?

On the other hand, when we move to a general decreasing monotonic transformation $Y=g(X)$, things become a little bit more complicated, making the general statement about switching between maxima and minima no so general, as recognized by Reviewer 1, who, unlike me, did not use 'straw man tactics', I hope.

In a nutshell, provided that X is Weibull, we already know the relationships between the distributions of X and Y in the case '$Y = 1/X$' and '$Y=-X$', and we already know that a general decreasing monotonic transformation $Y=g(X)$ cannot be used without a careful (case-by-case) check of the fulfillment of the (non negligible) hypothesis required to make the results meaningful.

Now, if this statement (or part of it, excluding '$Y=-X$') is the message of you paper, I simply cannot understand what is the content deserving a communication in a journal. By my side, I learned this stuff by reading books written 60 years ago. But reading old stuff is my perversion and fault.

That's all. If you, reviewers, and editorial board think that the message is worth a communication in HESS or wherever else, that's fine. I only highlighted that your message (and much more) can also be retrieved in good readings such as Gumbel (1958), books

from Kotz, Johnson and their colleagues, and other valuable texts. For me, spending time with those books/papers (i.e. getting in the giant's shoulders) is more enjoyable than writing papers.

I hope that this clarifies my position.

I wish you all the best

Sincerely

Francesco

PS: I always provide review reports as an eponymous reviewer (unless the Editors remove my signature due to journal's policies). So, if you did not agree with me in the previous review processes, you could have contacted me to discuss your point of view (regardless of Editor's decision), as we are doing now. I'm always open to exchange of opinions.

PS2: In the EMS review process, after recognizing your persistence in delivering your message, I suggested to the Editor to contact an additional reviewer, precisely a world-class expert in univariate and multivariate EVT (I do not mention his/her name here for the sake of correctness/privacy). So, we also agree on how a fair review process should proceed. I'm sorry that you did not have an additional opinion, but this did not depend on myself.

---

## Author Comment (AC1) · 13 Jul 2017

An immediate response is required here to the first part of the reviewer's comment, since it has implications for the review process. A final response will be added later covering other aspects, when the review deadline is up.

By way of a short clarification, the reviewer's reference to duality of maxima and minima is the sign connection between the asymptotic distributions of sample maxima and minima. So if X follows a distribution of maxima then Y follows a distribution of minima if Y = -X. This linkage, as noted by the reviewer, has been known for a very long time.

Because the reviewer has elected to publicly raise the history of this paper, the background needs to be set out a little more clearly. As the reviewer points out, the paper (essentially in its present form) was first submitted to *Advances in Water Resources*. It's not always easy to anticipate reviewer responses but I was astonished to see that I was charged with a rediscovery of the Y = -X extreme value relation. And that was it. The editor would not permit any author response. I had rediscovered Y = -X, the reviewer was of standing and therefore could not be questioned, so the paper had to be rejected.

That was a frustrating experience of course, but reviewers are busy people and can have bad days. In due course a new submission was made – to *Environmental Modelling & Software*. This time it was made very clear in the text that the paper had no connection with the Y = -X relation. Alas, it was to no avail. Once again the charge was made by the reviewer that the paper was a rediscovery Y = -X. And once again the paper was rejected, and once again no author response was permitted. I think if I could offer a single piece of advice to journal editors it would be: when faced an author and reviewer in direct conflict over basic content – seek a further independent reviewer.

The final step in the saga is the current submission to HESS. As part of the submission, an outline was provided to the Editors of the review history of the paper to date. In the manuscript I again tried to be as clear as possible over the Y = -X issue, noting both in the abstract and in the text of Section 2 that the proposed monotonic transformation in the paper …

**… would include, for example, $Y = X^{-1}$ but not $Y = -X$.**

I really don't think it could have been written more explicitly than that and I hoped that would be the end of the matter finally. Sadly, we are right down the same line as before, unbelievable as it may seem. So, for the third time, I stand accused of proposing a paper which in some way incorporates a rediscovery of the sign duality between extreme value maxima and minima.

There is, however, one important difference this time. Thanks to the HESS open review process, I at last have opportunity to respond on this topic and have been most pleased to do so here. The paper is there for all to see and anyone can verify that the reviewer criticism on the Y = -X issue is simply a straw man tactic. That is, the paper was deliberately defined to be something that it certainly never was, and then demolished on that basis. The extreme value Y = -X relation is in fact so totally unconnected to the paper that I'm surprised the reviewer did not go the whole way and deem the paper to be proposing a flat Earth as his basis for demolition.

As mentioned earlier, a final response to all reviewers will be made later. However, it is mentioned now that it is something of a low blow to both myself and HESS reviewers generally to imply that in some way I was using HESS to sneak a paper under the statistical review radar. Nor is the theme of the paper in any way related to presenting any new statistical distribution with much smoke and noise. I am in fact in full agreement with the reviewer that the "H distribution" is just one of an infinity of many, and will emphasise that point fully in a revised version. The inverse Weibull can be found in any text dealing with the Weibull distribution and its transformations – I should have made a reference.

In the meantime the review process can proceed to whatever conclusion. But can we please maintain the review standards expected of international scientific journals and not descend into straw man tactics.

Kind Regards    Earl Bardsley

---

## Author Comment (AC2) · 14 Jul 2017

My thanks for your comments Francesco

It's really good to be finally free of the criticism that I was rediscovering that a Weibull distribution could be created by a sign change applied to an EV3 distribution of maxima.

I won't comment more on the previous review history of the paper in the various journals. In my experience (and most others too) an editor rejection is pretty much a final thing and certainly is not the first step in a subsequent negotiation process with reviewers.

I guess for HESS it's something of an academic distinction between being a "reviewer" or "critical commentator". It's just easier for me just to use "reviewer". The Editor no doubt takes note of both reviews and any critical comment. Authors must meet (if they can) whatever objections are raised by both. Your points are certainly noted and will need to be responded to along with reviewer comments. However, as mentioned, I will leave all further responses on the technical stuff and the paper motivation until the end of the review process.

Kind Regards

Earl

---

## Short Comment (SC3) · 18 Jul 2017

F. Serinaldi

francesco.serinaldi@ncl.ac.uk

Dear Earl,

That's fine. I'm looking forward to a glass of wine ;-)

Concerning the H-distribution, even if I did not check carefully, it looks like the three-parameter Inverse Weibull mentioned e.g. here

Dragan Jukic D, Markovic D (2014) Total Least Squares Fitting the Three-Parameter Inverse Weibull Density, European Journal of Pure and Applied Mathematics, 7(3), 2014, 230-245 (http://ejpam.com/index.php/ejpam/article/view/2114)

The parameterization is different from yours, but it has a location parameter, which

seems to be the key additional parameter of your formulation. So, after reparameteri-
zation, they could match, I think. Anyway, this is only a very minor remark.

Sincerely,

F
* * *
198, 2017.

---

## Author Comment (AC3) · 18 Jul 2017

I am grateful to the reviewer for many helpful comments.

Yes, the assumption that a limiting Weibull distribution exists for the lower-bounded variable needs to be explicitly stated. As noted by the reviewer, there are situations where there will not be a limiting Weibull distribution for Y* even though a lower bound is present. In practice, however, the Weibull distribution is employed in numerous applied publications related to sample minima and the assumption must be made also in such cases that that the limiting distribution exists, usually supported empirically if a good Weibull fit is achieved to the sample minima.

The same assumption in applied work is required too when applying the GEV to sample maxima – that is, that a limiting distribution of maxima is assumed to exist. As with minima, this assumption of a limiting distribution of maxima can be shown to not hold in some situations. For example, Kotz and Nadarajah (2000, p. 55) cite Green (1976) as noting that maxima of random variables need not approach some stable limiting distribution. As mentioned by Kotz and Nadarajah (2000), the required assumption of a limiting stable distribution of maxima has not deterred researchers from applying the stable distributions in a multitude of applications.

In short then, the assumed Weibull limit distribution of Y* would seem to be no more or no less justifiable than the assumption of stable distributions of X* for sample maxima. Both approaches require distributions to be in the domain of attraction of extreme value distributions. This is the message of the submitted paper as a contribution to applied studies. There is no suggestion that any new extreme value theory is being introduced – it is simply a different approach for consideration for practical application by using existing theory in a somewhat different way.

The argument is that it is just as justifiable to obtain exceedance probabilities by using the Weibull distribution to estimate the probability that Y* is less than some value as it is to use the GEV to estimate the probability that X* is greater than some value. In practice the pragmatic justification would be fitting to data – fitting the X* values to the GEV or fitting Y* values to the Weibull distribution. This is the basis of the statement that the use of the Weibull distribution to obtain exceedance probabilities via Y* is no more or less valid than using the GEV to obtain exceedance probabilities via X*. However, the point of both approaches requiring stable distributions of extremes to exist could be made more clearly in any revision.

With respect to the second bullet point, there is certainly no suggestion that stable distributions of X* need imply stable distributions of Y*. But perhaps the converse might sometimes hold also. That is, situations might arise where there is a stable limit distribution of Y*, but not of X*.

It is only Y* that is needed in order to obtain an exceedance probability estimate via the Weibull distribution (provided there is a good data match and provided the assumption of a Weibull stable distribution of minima holds). However, practitioners have an understandable preference to visualise their degree of fit as measured against recorded maxima, rather than transformations of recorded maxima. This was the reason for introducing $Z = g^{-1}(W)$, because the degree of fit of recorded maxima to $f(Z)$ gives an implicit fit to the Weibull distribution also, and the fit to $f(Z)$ can be presented in the same scale as the original data.

With respect to the technical comments:

**P2 L8-12**  Yes.. the reference to the positive range of the variable is something of a distraction and is not an argument for the paper. It should just be mentioned in passing that the transformation approach yields distributions in the positive domain.

**P2 L19.** My apologies for being unclear in the text, which should have read: Let $f(x)$ be an unknown probability density function ..

**P2 L19-20.** Yes - text needs to be added to the effect that the Xi are assumed independent.

**P2 L21.**
**P3 L2**
My thanks for the clear definitions. The definitions would be added as indicated, in any revised version of the paper.

**P3 L3** Yes .. not all $g^{-1}$ will be applicable. If I may ask your help one further time, perhaps some wording could be suggested to define that only viable $g^{-1}$ should be employed?

Green, R.F. (1976) Partial attraction of maxima *J. Appl. Probab.* 13 159-163.

Kotz, S., and Nadarajah, S.: Extreme value distributions: theory and applications. Imperial College Press, London, 2000.

---

## Author Comment (AC4) · 18 Jul 2017

My thanks again for your comments Francesco – which I'm sure are well-intentioned.

We can agree at least that I was never seeking to rediscover that a Weibull distribution can be obtained as the negative of an EV3 random variable.

There was never any intention to suggest I was breaking new ground with a rediscovery of the inverse Weibull distribution. It was actually a question of terminology. Going by the definition in Prabhakar Murthy et al (2004, p. 115) the term "inverse Weibull" seems to be applied to the reciprocal of a random variable from a two-parameter Weibull distribution, which gives the inverse Weibull distribution as an EV2 distribution with location parameter zero. For the submitted paper, I needed a transformation applied to the three-parameter Weibull distribution with a non-negative location parameter. Because I was using a reciprocal transformation as an example I simply couldn't find a name for a distribution obtained as "the reciprocal of a random variable from a three-parameter Weibull distribution with non-zero location parameter" which was quite a mouthful. So I invented the term "H-distribution" as a convenient shorthand – nothing more than that. If that distribution has some existing name then I'm most happy to use the recognised name of course.

The reciprocal transformation was only used as the most simple example I could think of to illustrate the monotonic transformation. There is no suggestion that the H-distribution in itself is some form of discovery worthy of publication – though it does have interesting properties.

The value (or lack of value) of the submitted paper is based entirely on the suggestion that the monotonic transformation provides a basis for an alternative means, with extreme value justification, of estimating exceedance probabilities for positive-valued random variables, using the Weibull distribution as an alternative to the GEV. I believe that's a new suggestion. It is of course, subject to various specific assumptions which must be set out more clearly than I did in the original paper – as was noted by the reviewer. It may happen in fact that the proposed monotonic transformation is not viable from a math viewpoint and that would be sudden death for the whole concept. That's something for the reviewer to make comment on and I'm happy to go with whatever the decision is. Please see also my responses to the reviewer comments.

Just by way of passing mention, I too have Gumbel's text on my shelf – ever since 1980 in fact when I purchased it (along with some others) while I was completing my PhD on applications of extreme value theory in the Earth sciences.

Irrespective of the final review outcome, I'm sure we could enjoy a glass of wine together if our paths should cross at a conference somewhere.

Best wishes

Earl

Prabhakar Murthy et al 2004. Weibull Models. Wiley-Interscience

---

## Referee Comment (RC2) · Anonymous Referee #2 · 22 Dec 2017

After reading this technical note I wonder why the Author did choose HESS and why He does not try to demonstrate a bit more why hydrologists or earth system scientists should be interested in His results. I think F. Serinaldi is right when he says that "proposing 'new' statistical results in hydrologic journals is not a good strategy to guarantee the quality of the scientific production, as there is a very good chance that professional statisticians with expertise in EVT do not comment on the paper". I am not an expert in statistics but a user of statistic for hydrologic applications and I would have expected to get some insight on how to use these new distributions in hydrology from a paper in HESS. Personally I like the idea of non-negative extreme value distributions for the cases in which Monte Carlo simulations are needed and one does not want to generate negative flood peaks, for example. However I do not think that having

an upper bound is in general a good idea in hydrology (see e.g. Papalexiou and Koutsoyiannis, 2006). I would suggest the Author either to submit His work to a statistical journal or to extend it in a way to make it useful for the readership of HESS, which would imply to demonstrate the applicability of the method in hydrology, for example by showing that the proposed distribution is more appropriate that the standard ones in a particular case study.

Minor comments:

- The abstract, which includes definitions and one equation, is very unusual for hydrology journals. Besides, Section 2 is a repetition of the abstract. I would suggest to shorten it.

- Figure 2: I would suggest to plot all three distributions of Figure 1 also here

Citations:

Papalexiou, S. M. and D. Koutsoyiannis (2006) A probabilistic approach to the concept of Probable Maximum Precipitation. Advances in Geosciences, 7, pp.51-54.

---

## Author Comment (AC5) · 4 Jan 2018

My thanks to Referee # 2 for comments, and also to Referee #1 as well as Francesco Serinaldi for independent comment.

Responses to Referee # 1 have already been posted, but a further comment is added here with respect to the final Referee #1 point raised:

*If Y\* has a Weibull distribution then it is not guaranteed that $g^{-1}$ is defined on
the whole of the range of Y\* which always runs up to + ∞.*

In response, the definition of the transformation $g(X)$ should be extended to read:

"Define the transformation $Y_i = g(X_i)$ where $g(X)$ is a positive strictly monotonic decreasing function of $X$. Further, let the inverse function $g^{-1}(X)$ be defined over the same range as $X$."

This definition is also incorporated in response to Francesco Serinaldi's comment on the need for clear definition of $g(X)$.

Responses to Referee # 2 follow below.

*I wonder why the Author did choose HESS and why He does not try to demonstrate a bit more why hydrologists or earth system scientists should be interested in His results. I think F. Serinaldi is right when he says that "proposing 'new' statistical results in hydrologic journals is not a good strategy to guarantee the quality of the scientific production, as there is a very good chance that professional statisticians with expertise in EVT do not comment on the paper".*

HESS was chosen because the paper is intended for hydrological application and HESS is a well-known hydrological journal with an open review process. The latter is important because I felt it would be helpful in this instance to have an open discussion as part of the review.

I have made response previously with respect to the Serinaldi comment on "new" statistical methods. I could perhaps add that if I felt that I really had contributed some new aspect of extreme value theory then of course submission to a specialist journal such as *Extremes* would have been appropriate. However, no doubt Reviewer # 1 can confirm that the paper involves no new theory worthy of such high-level consideration. Indeed, the basic idea is just a generalisation of the concept " for positive-valued data you can estimate the probability of getting a larger value by estimating the probability of getting a smaller value of the reciprocal". The paper nonetheless does have something useful to offer for practical application in that for sufficiently large sample sizes it is pointed out that the Weibull distribution (as asymptotic distribution of smallest extremes) can be used as an alternative to the GEV as a single expression for estimating exceedance probabilities for positive-valued variables, and with no less extreme value justification. This is conditional on existence of extreme value limit distributions in any given instance (as noted by Referee #1), but these conditions apply as much to distributions of maxima as to distributions of minima. Because hydrologists will probably prefer working in terms of the original measurements rather than transformed data, I have suggested transforming the limit Weibull distribution to create new distributions for application to maxima, presented in the original scales. However, while perhaps useful for hydrological data representation, such transformations cannot be proposed as a statistical advance.

*I would have expected to get some insight on how to use these new distributions in hydrology
from a paper in HESS.*

My thanks for the suggestion. Yes - the paper would benefit from some additional comment in this regard. In fact, the advantage of the proposed approach is that is that H distribution, and the many other possible distributions of maxima in the class as defined, are all derived from transformations of Weibull distributions with non-negative values of the location parameter. Therefore, no new formal estimation methods are required for practical application beyond those already developed for 3-parameter Weibull distributions.

Informal Weibull estimation by curve fitting could also be used along the lines suggested by Bardsley (1989. This text would be added to a revised paper together with an example application.

*Personally I like the idea of non-negative extreme value distributions for the cases in which Monte Carlo simulations are needed and one does not want to generate negative flood peaks, for example.*

I didn't have simulation in mind but it would be nice if the distributions found some application for simulation purposes in this way.

*However I do not think that having an upper bound is in general a good idea in hydrology (see e.g. Papalexiou and Koutsoyiannis, 2006).*

Following the cited reference, this referee comment is probably more with respect to the practicalities of defining and estimating an upper bound, particularly in the context of probable maximum precipitation. The limits of physical processes ensure that environmental variables must indeed have upper bounds, poorly defined though they may be. We can always specify some impossibly large magnitude to make the point. For example, a 10,000-metre daily rainfall anywhere on Earth does not have a vanishingly small exceedance probability, because the exceedance probability is exactly zero.  The same applies for a 1,000-metre daily rainfall.
So somewhere below 1,000 metres of rain there must be a universal upper bound to daily rainfall amounts. To be sure, this bound is so nebulous that it is best expressed as a probability distribution. However, that distribution itself must be bounded above and does not extend to infinity. There is nothing new about extreme value distributions bounded above – as is the case with the Type 3 extreme value distribution of largest extremes. Having apparent Type 1 or Type 2 distributions which are mathematically bounded above is in fact consistent with the reality of upper bounds for environmental variables. That is, having data well-matched by Type 1 or Type 2 extreme value distributions is not evidence to negate the existence of upper bounds.
The important practical point to make, however, is that any estimated value of an upper bound in such situations will be so poorly defined by the data as to be meaningless and will have little effect on estimated exceedance probabilities over the data range and for some distance beyond the largest value. Put another way, it would take a very large number of simulations for the upper bound to have any evident effect in distribution (b) in Figure 1.
With the provisos mentioned, incorporating upper bounds in hydrological parameterisations is more a good idea than a bad idea.

*I would suggest the Author either to submit His work to a statistical journal or to extend it in a way to make it useful for the readership of HESS, which would imply to demonstrate the applicability of the method in hydrology, for example by showing that the proposed distribution is more appropriate that the standard ones in a particular case study.*

As noted, the paper would not be acceptable as a contribution to a statistical journal. For demonstrating usefulness for HESS readers, probably the best way would be to use some data simulated from a rescaled version of the present Figure 2 distribution – which is not well described by any GEV.

*The abstract, which includes definitions and one equation, is very unusual for hydrology journals. Besides, Section 2 is a repetition of the abstract. I would suggest to shorten it.*

Yes – the abstract should be reduced and made less explicitly mathematical. I also need to add "Technical Note: " in the title.

*Figure 2: I would suggest to plot all three distributions of Figure 1 also here*

Yes – that would be better for completeness, my thanks for noting it.